# Diagnostic Practices and Treatment for *P. vivax* in the InterEthnic Therapeutic Encounter of South-Central Vietnam: A Mixed-Methods Study

**DOI:** 10.3390/pathogens10010026

**Published:** 2020-12-31

**Authors:** Thuan Thi Nguyen, Xa Xuan Nguyen, Maya Ronse, Quynh Truc Nguyen, Phuc Quang Ho, Duong Thanh Tran, Rene Gerrets, Kamala Thriemer, Benedikt Ley, Jutta Marfurt, Ric N. Price, Koen Peeters Grietens, Charlotte Gryseels

**Affiliations:** 1National Institute of Malariology, Parasitology and Entomology (NIMPE), 34 Trung Van, Trung Van Ward, Nam Tu Liem District, Hanoi 10000, Vietnam; xanguyenxuan60@gmail.com (X.X.N.); trquynh258@gmail.com (Q.T.N.); hoqphuc@gmail.com (P.Q.H.); tranthanhduong@hotmail.com (D.T.T.); 2Medical Anthropology Unit, Department of Public Health, Institute of Tropical Medicine, Nationalestraat 155, 2000 Antwerp, Belgium; mronse@itg.be (M.R.); kpeeters@itg.be (K.P.G.); cgryseels@itg.be (C.G.); 3Amsterdam Institute for Social Science Research (AISSR), University of Amsterdam, Postbus 15718, 1001 NE Amsterdam, The Netherlands; rgerretswork@outlook.com; 4Amsterdam Institute for Global Health and Development (AIGHD), AHTC, Tower C4, Paasheuvelweg 25, 1105 BP Amsterdam, The Netherlands; 5Global Health Division, Menzies School of Health Research and Charles Darwin University, Rocklands Drive Casuarina, Darwin Northern Territory 0810, Australia; kamala.ley-thriemer@menzies.edu.au (K.T.); benedikt.ley@menzies.edu.au (B.L.); jutta.marfurt@menzies.edu.au (J.M.); ric.price@menzies.edu.au (R.N.P.); 6Centre for Tropical Medicine, Nuffield Department of Clinical Medicine, University of Oxford, Oxford OX3 7LG, UK; 7Mahidol-Oxford Tropical Medicine Research Unit (MORU), Faculty of Tropical Medicine, Mahidol University, Bangkok 10400, Thailand

**Keywords:** malaria elimination, G6PD deficiency, *P. vivax*, acceptability of interventions, ethnic minorities, Vietnam

## Abstract

Malaria elimination in the Greater Mekong Sub-Region is challenged by a rising proportion of malaria attributable to *P. vivax.* Primaquine (PQ) is effective in eliminating the parasite’s dormant liver stages and can prevent relapsing infections, but it induces severe haemolysis in patients with Glucose-6-Phosphate Dehydrogenase (G6PD) deficiency, highlighting the importance of testing enzyme activity prior to treatment. A mixed-method study was conducted in south-central Vietnam to explore the factors that affect acceptability of G6PD testing, treatment-seeking behaviors, and adherence to current regimens. The majority of respondents (75.7%) were unaware of the different parasite species and rather differentiated malaria by perceived severity. People sought a diagnosis if suspected of malaria fever but not if they perceived their fevers as mild. Most respondents agreed to take prescribed medication to treat asymptomatic infection (94.1%) and to continue medication even if they felt better (91.5%). Health professionals did not have G6PD diagnostic tools nor the means to prescribe PQ safely. Adherence to treatment was linked to trust in public providers, who were perceived to make therapeutic decisions in the interest of the patient. Greater focus on providing acceptable ways of assessing G6PD deficiency will be needed to ensure the timely elimination of malaria in Vietnam.

## 1. Introduction

*Plasmodium falciparum* (*P. falciparum*) and *Plasmodium vivax* (*P. vivax*) are the dominant parasite species causing malaria in humans. Outside of sub-Saharan Africa, a rising proportion of malaria is attributable to *P. vivax*, with the majority of cases concentrated in the Asia-Pacific region [1,2,3]. Despite the perception that *P. vivax* causes a more benign infection than *P. falciparum*, its ability to form dormant liver stages that result in the recurrent episodes of malaria adds significantly to its morbidity and mortality, particularly in remote and socio-economically deprived communities [4,5,6,7]. 

Six countries in the Greater Mekong Sub-Region (GMS), including Cambodia, Yunnan province, and the Guangxi Zhuang Autonomous Region of China, Laos, Myanmar, Thailand, and Vietnam, have committed to eliminate malaria by 2030 [8,9]. Regional data from 2010 to 2018 demonstrate an overall reduction in malaria incidence by 76% [1]. However, this is mostly due to a decline in the incidence of *P. falciparum*, whereas there has been an increase in the proportion of *P. vivax* in most countries [10,11,12]. In Vietnam, from 2006 to 2010, *P. falciparum* still made up 70% of all confirmed malaria cases [13]. From 2011 to 2019, *P. vivax* has become increasingly dominant, accounting for almost 40% of all confirmed cases with a high concentration of cases in nine provinces in the south-central region [14]. In these pre-elimination settings, tackling *P. vivax* is crucial for the regional and national malaria elimination agenda. 

*P. vivax* forms dormant liver stages (hypnozoites) that are asymptomatic and undetectable from peripheral blood examination. Hypnozoites can reactivate, causing recurrent episodes of infection (relapses) weeks to months after the initial infection [15,16,17,18,19]. Currently, primaquine (PQ) is the only effective and widely available drug to kill the hypnozoite stages [20,21]. In areas with chloroquine-sensitive *P. vivax*, including the GMS countries, the WHO’s treatment guidelines stipulate that in addition to chloroquine (CQ), a 14-day course of PQ is needed to kill the hypnozoites and prevent relapses (“radical cure”). The prolonged treatment regimen challenges patient adherence, leading to poor effectiveness [22,23]. Even though PQ is generally well tolerated, it can cause severe haemolysis in individuals with low activities of the glucose-6-phosphate-dehydrogenase (G6PD) enzyme [24,25,26,27,28]. Therefore, the WHO has recommended that national malaria control programs (NMCPs) should adopt G6PD testing prior to rolling out a PQ-based radical cure [29]. The current gold standard for testing G6PD deficiency is a quantitative measurement using UV spectrophotometry [30,31]; however, for field deployment, qualitative or semi-quantitative methods are more commonly used [32]. 

In line with the WHO’s treatment guidelines, the NMCP in Vietnam recommends the use of a PQ-based radical cure for patients with uncomplicated *P. vivax and P. ovale malaria* [33]. Whilst the national treatment guidelines do not recommend G6PD testing prior to administering PQ, they recommended patients with significant haemolysis following PQ, although do not specify the testing method. The availability of G6PD deficiency diagnostics is limited to a number of specialized hospitals at national, regional, and provincial levels (with testing methods varied, including both quantitative and qualitative) but unavailable at district or commune level. Hence, most endemic provinces have not yet acquired the necessary diagnostic capacity or equipment to provide this additional health service, and this undermines the safe and effective treatment of *P. vivax* malaria. 

The Vietnamese National Institute of Malariology, Parasitology and Entomology (NIMPE) joined the Asia Pacific Malaria Elimination Network (APMEN) in 2009 to review the challenges and explore solutions for implementing radical cure and point-of-care G6PD testing effectively together with research partners and other NMCPs in the region [26,34]. In the GMS, the remaining malaria hotspots are mainly concentrated amongst hard-to-reach ethnic minorities and mobile migrant populations whose limited uptake of preventive measures and treatment presents an additional difficulty for the efficient roll-out of G6PD testing and radical cure [35,36,37,38,39]. The south-central highland province of Ninh Thuan is largely populated by people of Ra-glai ethnicity. Malaria transmission persists in this region despite the implementation of malaria elimination interventions [36,40,41,42,43]. 

As part of a multi-center project across the region, a mixed-method study was conducted from 2018 to 2019 in a malaria-endemic district of Ninh Thuan to explore the acceptability of radical cure and G6PD testing amongst Ra-glai community members and their healthcare professionals. Specifically, the study examined factors of importance for the implementation of the novel G6PD diagnostic tool including local people’s perceptions, health-seeking behaviors, and treatment adherence related to malaria.

## 2. Materials and Methods 

### 2.1. Study Site and Population

The study was conducted in seven villages in the mountainous district of Bac Ai in Ninh Thuan province (Figure 1). Bac Ai has the highest malaria incidence in Ninh Thuan [44]. Malaria cases usually peak twice a year, from April to July and September to November, which coincide with the rainy season. Surveillance reports show a substantial reduction in malaria cases in the last ten years, from 1623 cases in 2009 to 53 cases in 2019 [45]. 

At the time this study was conducted, malaria diagnostics used by the local health services included finger prick sampling, microscopic examination of giemsa stained blood films, and a conventional rapid diagnostic test (RDT). Although routine G6PD testing was not part of the routine healthcare, some health centers in Bac Ai district participated in G6PD research studies during the study period using the qualitative fluorescent spot test (FST) [46]. Previous malaria studies conducted in the region reported no challenges on blood sampling and the acceptance of malaria diagnostic tools amongst the local population [46,47,48]. 

In 2018, the district had a population of approximately 28,000 inhabitants with more than 90% belonging to the Ra-glai ethnicity [49], the rest mostly being Kinh, which is the dominant ethnicity in Vietnam. Ra-glai people speak a language from the Malayo-Polynesian language family, and no standard writing existed at the time of this study. In the past, they were scattered in the highlands of their traditional territory, which was referred to as “the old village”, and they relied on swidden agriculture, hunting, and gathering [36,40]. From the 1990s to 2000s, the government resettled the Ra-glai in government-constructed villages in the low lands, which was referred to as “the new village”, in an effort to abolish their slash-and-burn farming practice and integrate them economically and culturally into Vietnamese society [50]. Most of Ra-glai traditional farming land and territory were turned into either protected forest reserves or hydropower dams. Many Ra-glai have lived in poverty and continued to grow crops in the mountainous locations that are close to the forest and belong to their traditional highland territory [43,51]. 

### 2.2. Study Design 

We conducted an exploratory sequential mixed-method study, triangulating qualitative and quantitative strands [53,54]. The study was embedded in a larger health facility-based study that was undertaken to determine the populations at risk of PQ-induced hemolysis using novel G6PD diagnostic tools. The qualitative strands aimed to provide an in-depth understanding of the local context and associated health-seeking behaviors. Findings from the first qualitative study were used to inform the design of a structured questionnaire in the quantitative strand. The quantitative strands aimed to determine the populations at risk of malaria and PQ-induced hemolysis and consisted of a community-based cross-sectional survey and a health facility-based survey enrolling patients for blood sampling (results will be presented separately). In the course of the community-based survey, a social science questionnaire was administered to assess respondents’ knowledge and perceptions of malaria, health-seeking behaviors, and adherence to treatment. An additional short questionnaire was embedded in the health facility-based survey to measure the patient’s perceptions of *P. vivax*. After both quantitative surveys, a second strand of qualitative research was conducted to triangulate and enrich the quantitative findings. 

#### 2.2.1. Qualitative Strands

*Data collection*. The qualitative data collection tools were developed and refined following an iterative process with the aim to reach saturation [55]. In the beginning of the study, a topic guide was developed containing open-ended questions about the local context and people’s experiences in and perceptions of different aspects of malaria health services. New questions and themes that emerged from the data were gradually added to the topic guide. Data were analyzed intermittently during fieldwork. 

Qualitative data were collected in seven villages during three periods of fieldwork from May 2018 to July 2019 (each period lasted four to six weeks). Data collection techniques included informal conversations (IC), in-depth interviews (IDI), group discussions (GD), and participant observation (PO). After obtaining oral consent from the participants, the researcher proceeded to interviewing in either Vietnamese or in Ra-glai language, with the help of a translator when required. The content of interviews was written down on fieldnotes during the conversation. When the circumstances permitted and in agreement with the interviewee, interviews were audio recorded. 

*Sampling*. Purposive sampling was applied. The researcher developed rapport with community members and key informants through staying in different villages. Key informants knowledgeable of the socio-cultural context and the local health services including malaria diagnosis and treatment were targeted, including public healthcare professionals, private and informal health service providers, and village leaders. The sampling strategy aimed at having diversity in community members from both the “old” and “new village” with diverse socio-economic status and occupations. 

*Data analysis*. The researcher consistently tested and validated hypotheses in the field with the existing theory as they were generated from preliminary findings. An inductive approach was applied to qualitative data analysis using NVivo 12 (QSR International Pty Ltd., Cardigan, UK). Collected data were organized in codes and categories. Codes that referred to common concepts were grouped together. These codes and categories were arranged in the form of a tree diagram that served as an analytical frame for the qualitative study. 

#### 2.2.2. Quantitative Strand

*Data collection.* A paper-based social science questionnaire was administered from May to June 2019 as part of the community-based cross-sectional survey. The questionnaire consisted of 28 items to quantify demographic characteristics, perceived health status, malaria perceptions, health-seeking behaviors, and perceptions of fever. The questionnaire was first developed in English and then translated to Kinh (in writing) and then to Ra-glai (verbally). Preliminary qualitative findings were used to continuously adapt the questionnaire during two periods of fieldwork before backward translation and finalization in Vietnamese and English. In the health facility-based survey, a questionnaire was administered to enrolled febrile patients (or their guardians) who sought treatment at public health facilities from May to November 2019. This short questionnaire assessed the patient’s understanding of the different types of malaria and reasons for visiting the health center. 

*Sampling.* In the cross-sectional survey, 892 households from four villages (based on the local census data of Bac Ai) were used as the sampling frame, of which 202 were selected randomly using a computer program (http://www.random.org). They were tested for malaria using conventional RDT (RDT SD Bioline Malaria Ag P.f/Pan, Abbott, Chicago, IL, USA) as approved and used by the NMCP. Whenever a household refused to participate, refusal was recorded, and replacement was made by a randomly selected alternative household. The study collected ≤7.5 mL venous blood from one randomly selected household member and ≤400 μL capillary blood each of the rest of the household members above the age of one year (data on venous participants will be presented separately). Basic demographic data, information on past malaria episodes, and mobility were recorded for each household member, whilst the individual who agreed to give venous samples answered the social science questionnaire. In the additional health facility-based survey, all patients who were older than 12 months of age and reported to have had a fever 48 h prior to arrival at the health facility were invited to participate. Recruited participants for venous sampling in the community-based survey and patients in the health facility-based survey were tested for malaria (using conventional and ultra-sensitive RDT) and G6PD deficiency (using the novel quantitative diagnostic tool G6PD RDTs Accessbio/Carestart, Somerset, NJ, USA). 

*Analysis*. Questionnaires were entered in EpiData version 4.4.2.1 (The EpiData Association, Odense, Denmark) and then exported to Stata 12.1 (StataCorp LLC, College Station, TX, USA) for data cleaning and analysis. Descriptive statistics were used to indicate proportions and frequencies of key variables. 

### 2.3. Ethical Considerations

The study was implemented by NIMPE in collaboration with the Institute of Tropical Medicine in Antwerp (ITM), Belgium, and the Menzies School of Health Research, Australia. Ethical clearance for the study was granted by the Ministry of Health and NIMPE Ethical Review Board (decision 1648/QD-VSR), ITM Institutional Review Board (IRB/AB/AC/040), and the Human Research Ethics Committee of the Northern Territory (HREC: 2017-3005). 

In the qualitative strand, the researcher sought verbal consent from informants [56] based on previous work in the study population that indicated that illiteracy and mistrust toward outsiders taking notes or signing documents were high. In addition to gaining individual oral consent, the researcher also sensitized the local authorities and traditional leaders in the study villages about the study to support acceptance and trust building in the study villages. 

In the quantitative strand, written informed consent was collected from the participant or the legal guardian with patients who were under 18 years of age. Trained fieldworkers, who were fluent in Ra-glai language, were selected to administer consent procedures and the questionnaire. At first, they gave verbal explanation in Ra-glai language about the study and its eligibility to the respondent. If the study participant could not read nor write, a thumbprint was collected. 

## 3. Results

### 3.1. Respondent Characteristics

In the qualitative strand, a total of seven group discussions, three group interviews, 54 informal conversations, 47 in-depth interviews, and 44 participant observations were conducted. Included in the sample were community members of different age groups living in both the “old” and the “new village”, shamans, local leaders, nurses, medical doctors, health program administrators, managers, pharmacists, laboratory technicians, and shop owners. 

In the community-based cross-sectional survey, 202 household representatives participated. There were 26 households who were initially selected but did not participate because of refusal to give blood samples (n = 17) and absence from home (n = 9). In the health facility-based survey, 176 patients were eligible and gave consent to participate. The characteristics of survey respondents from both quantitative strands are reported in Table 1.

### 3.2. Community Perceptions of P. vivax Malaria and Its Treatment

#### 3.2.1. Perceived Malaria Types

In the cross-sectional survey, 75.7% (153/202) of respondents did not know about different malaria species and therefore could not distinguish between *P. vivax* and *P. falciparum* malaria. Overall, 29.7% (60/202) of respondents stated an understanding about the occurrence of relapses and 52.0% (105/202) of respondents did not think a person could have malaria without having symptoms (Table 2). In general, malaria was considered a common type of fever in the community by 31.7% (64/202) of respondents (Table 3). Among the 176 patients enrolled in the health-facility based survey, only 5.7% (10/176) had ever heard about the two most common malaria species in the area, *P. vivax* and *P. falciparum* (Table 2). Only 4.0% (7/176) of patients were aware of the different symptoms caused by each type of malaria, with 1.7% (3/176) believing that *P. falciparum* caused more severe symptoms. A total of 4.6% (8/176) of enrolled patients in the health-facility based study had a positive malaria test, according to the conventional RDT, of which two infections were confirmed by microscopy to be *P. vivax* (2/8, 25.0% of total number of malaria cases). Prior to their diagnosis, 11.9% (21/176) of patients suspected their fever was caused by malaria. Further analysis showed that none (0/8) of the malaria-positive patients suspected that they had malaria prior to testing. 

The information on the two main malaria species in the region was not featured in the printed health materials about malaria available at public health centers. At the time of the study, health materials placed a strong focus on the risk and the link between mosquito bites, malaria, and forest exposure. Health workers had some knowledge of falciparum and vivax malaria but thought that the biomedical specifications about malaria species did not match Ra-glai illness perceptions. Informants perceived the explanation on different malaria species toward patients to be difficult as well as unnecessary. Ra-glai were perceived by health staff as unable to understand the differences between malaria species nor its implications for medical procedures. 

Ra-glai informants explained that they embodied the severity of malaria symptoms (not necessarily followed the medical classification of malaria) and differentiated between two types of malaria based on perceived severity: *“sốt rét thường”* referred to “normal malaria fever” and *“sốt rét nặng”* referred to “severe malaria fever”. Community members described “normal malaria fever” as an illness with mild fever and no major disruption to the patient’s sleep, appetite, and moving or working capacity. “Severe malaria fever” meant the patient’s bodily functions and daily life were significantly disrupted by severe symptoms such as convulsion and unconsciousness. During consultations, prescribers sometimes indicated “normal malaria” to the patient with the intention to reassure the patient of a normal illness progression and that recovery was expected within a few days. When someone was diagnosed with “severe malaria”, the prescriber sometimes suggested that the patient take supplement treatment such as vitamins in intravenous (IV) drips and rehydration solutions to boost the healing process (see below).

#### 3.2.2. Perceived Malaria Aetiology

In the community-based survey, an open question gauged the respondent’s perception of the causes of malaria first, after which qualitatively identified relevant options were actively prompted by the interviewer. Both through open and prompted answers, the majority of the respondents were aware of the link between mosquito bites and malaria (73.3% (148/202) before prompting and 89.6% (181/2020) after prompting (Table 3). Although “living in the old village” was not selected by respondents as a main cause when the question was left open (21.8%, 44//202), after prompting this option, the majority of respondents confirmed this as one of the potential aetiologies of malaria (77.7%, 157/202). Social desirability bias might play a role in this response rate, as the qualitative strand found that discouraging people to move back to their “old village” was used as a strategy both by health workers and the local government to promote adherence to state programs that aimed to eradicate both slash-and-burn agriculture and the associated forest malaria. 

#### 3.2.3. Perceptions of Appropriate Malaria Treatment

In total, 94.1% (190/202) of respondents in the community-based survey stated they would take malaria medication when prescribed even if they did not have symptoms of malaria (Table 4), and 91.5% (185/202) of respondents said that patients with malaria should continue taking all medicines even if they felt better. Social desirability bias might have inflated the response regarding the acceptance of antimalarials due to the perception that this study was part of the government’s malaria elimination interventions. In addition, local healthcare workers were part of the research team, and the recognition that antimalarials were only dispensed by these public health providers may have further impacted people’s responses. Indeed, all respondents who self-reported having had malaria in the past also reported to have sought treatment from the community health center (CHC) (Table 4). Adhering to a doctor’s prescription at public health facilities was a way for the patient to continue having access to free-of-charge medical service. However, when fever symptoms or signs of sickness recurred, it was common for patients to seek new treatment from a different provider as they considered the treatment of the first provider insufficient to cure the illness. 

Interviews with individuals who had previously had malaria found that people were unaware of the name and purpose of most antimalarial tablets. Only 19.0% (4/21) self-reported on having taken medication for more than three days, indicating that they were either not prescribed PQ-based radical cure or did not adhere to the treatment. Interviews in the qualitative strand found that patients actively sought treatment for all kinds of febrile illnesses at both private and public providers. If they suspected malaria causing their fever, they actively sought diagnosis at the CHC or the district health center (DHC) in order to receive antimalarials. Informants shared that they knew over-the-counter (OTC) medicines for fevers could not treat malaria, and antimalarials were not provided by private providers. Some patients perceived the purpose of antimalarials was to treat fevers, so when the fever was diminished, they stopped taking the rest of the antimalarials. For perceived mild or non-malaria fevers, people did not go to the CHC or DHC for diagnosis and treatment. They said the medicine to treat non-malaria fevers provided at these public providers was taking longer to treat symptoms, which risked the relapse of symptoms. Non-malarial fevers were thought to be best dealt with by either waiting a few days to see if symptoms disappeared or with OTC medicine from pharmacies or private providers. Despite providing treatment for fevers, private providers are not allowed to sell or administer malaria diagnostic tests. Patients with mild fevers who initially sought OTC treatment only resorted to malaria diagnostics at CHC or DHC when OTC medicines were not proving effective to stop the fever. 

More than half (66.7%, 14/21) of respondents stated they used additional treatments besides antimalarials provided by the CHC (Table 4). These included intravenous drips (92.9%, 13/14), injection (7.1%, 1/14), and spiritual sacrifice (14.3%, 2/14). Informants explained intravenous (IV) drips usually included rehydration fluids and vitamins, while injection could include antibiotics. These medical supplements and rituals (to appease their ancestors) were taken to boost the healing process. Even though the additional medical treatments were paid out-of-pocket by the patient, these were perceived as an important part of the total package of care needed to tackle the illness. To some patients, having the option to rest at the health facility whilst receiving IV drips instead of directly going home after the consultation was perceived as speeding up recovery.

“People throw away their medicine [antimalarials] when they see the symptoms have reduced. Or they save these pills for next time they get fever… Not everyone can understand well the instructions in Kinh language at the CHC… Patients here do not have the custom of asking their doctor for the cause of their illness. They only describe the symptoms to the prescribers.” (IC, a female villager, farmer).

### 3.3. G6PD Testing and Malaria Diagnostics 

#### 3.3.1. Healthcare Staff Perceptions of G6PD Testing

Prior to the health facility-based study, the DHC had better capacity to handle different diagnostic tools, including their past experience in implementing the qualitative FST tests [46]. In this study, blood samples were collected by the CHC health worker and then transported to and processed by the laboratory at the DHC. Laboratory technicians working at district and provincial level perceived the quantitative G6PD RDTs testing evaluated by this study [57] as a new and easy-to-do diagnostic tool. However, general healthcare workers lacked information about the importance of G6PD deficiency testing and its role in PQ prescription. Interviewed healthcare workers perceived the work division for G6PD testing between the CHC and the DHC (as implemented in the study) as time-consuming and pointed to the importance of making G6PD testing available at not only the district but also the community level to avoid unnecessary workload for both CHCs and DHCs (e.g., transportation of samples and results) and possibly a longer waiting time for the patient. However, informants emphasized the importance of providing the CHC with sufficient human and technological resources to achieve this, as they were already over-burdened by their routine work. As an illustration, according to the local health registry, the CHC in Phuoc Thang, which consisted of five health workers, provided primary health services to approximately 500 patients each month for a population of 4000, with fevers as one of the top three reasons why people sought care. 

In the routine health service from the community to provincial level, health staff relied on malaria RDT and microscopic screening (MS) as the only two malaria diagnostic tools, despite its shortcomings for the diagnosis of *P. vivax*. Molecular diagnosis and G6PD deficiency testing using the UV spectrophotometry and the FST tests were not provided as part of the free-of-charge diagnostics. Interviewed program managers perceived the main challenges for the provision of G6PD testing in routine care to be the lack of critical information about the accuracy and cost-effectiveness of point-of-care G6PD diagnostic tools. Another important challenge, according to health staff, was insufficient training for laboratory technicians particularly at commune level and the lack of adequate laboratory equipment in both the CHC and the DHC. Some informants mentioned they felt it necessary to revise the national guidelines for malaria diagnosis and treatment to explicitly include G6PD diagnostic testing in routine malaria service. 

#### 3.3.2. Healthcare Staff Diagnostic Practices for Malaria

Generally, health staff in CHCs used malaria RDTs as the primary diagnostic tool for malaria, and they usually communicated the results to the patient within 15–20 min. Microscopy was perceived as a secondary diagnostic method because the one person at the CHC who could perform this task often had difficulties in making time for this activity within working hours. If microscopy was not done on the same day, health staff said that they informed patients by phone when the result was available. Health staff reported the difficulty in getting hold of patients as they went back to their farms with no mobile phone reception. Health staff sometimes expressed their frustration of the accumulated microscopic workload, whilst financial incentives to perform such tasks were not provided. The DHC was reportedly available to assist with MS in case the CHC did not have the capacity to do so. In practice, the DHC’s workload was often also too high to accommodate the request to perform microscopy for regular malaria diagnosis, let alone assist with additional tasks such as G6PD testing. In this study, most of the G6PD tests and analyses in the clinical study were performed by the field workers from NIMPE as a solution to overcome the challenge of the high workload at health facilities at commune and district level. 

#### 3.3.3. Patient Perceptions of Malaria Diagnostics 

Informants in the qualitative study explained that if they suspected malaria, for instance because they were having severe symptoms such as sequential chills and high fever, vomiting, fainting and/or convulsions, they would definitely seek diagnosis at the CHC or the DHC. People knew malaria diagnostics were only available from these public providers and not from private practitioners or pharmacies. However, when they had mild symptoms, e.g., mild fever, headache, fatigue—which could potentially indicate infection with *P. vivax*—they did not go to a CHC or DHC for diagnosis and sought medicines to treat their fevers at home or from informal pharmacies and practitioners. Seeking diagnosis was usually delayed until symptoms worsened. Of the 10.4% (21/202) of community-based survey respondents who self-reported to ever have had malaria (Table 4), the median time at which they reported to have sought diagnosis was three days (ranging from one to four days) from the onset of symptoms (Table 5). All malaria patients (21/21) reported having gone to a CHC for treatment and received diagnosis using a malaria RDT. More than half (12/21) of them stated that their blood samples were analyzed using microscopy and that was associated with a longer time to receive results (Table 5). From the qualitative interviews, it was clear that trust in the results of conventional RDTs and microscopy was absolute, even amongst health staff. Therefore, patients accepted a finger prick for RDT and microscopy if prescribed, even though most individuals did not have good knowledge of microscopy and what it did in terms of diagnosis. 

Health workers did not explain the risk of anemia to patients when taking certain antimalarials nor how G6PD testing could help improve the safety of vivax treatment. Additional blood sampling (i.e., venous blood for G6PD testing), which was required as part of the health facility-based survey, was perceived as invasive and detrimental to the well-being and health of people, as it was thought to delay healing and impact on their ability to work the fields. Informants provided accounts of experiences with venous blood sampling prior to this study and explained how they perceived it as an imposed medical procedure that was disruptive to their ability to carry out farming work. Several key informants described how blood was considered an important element of good health and essential to the strength required for farming work. People associated losing blood with losing a person’s strength and increasing one’s vulnerability. Children, women, and elderly were often perceived as weak, fragile, and ineligible to give more blood than the few drops required for finger-prick malaria testing. Informants explained that it took nutritious food, which was often scarce in Ra-glai families, to produce more or new blood in their body. Informants believed malaria testing and venous blood sampling were only justifiable for individuals who had a significant risk of contracting malaria, such as those staying overnight in the forest. 

Reservations toward the additional blood sampling needed for the G6PD test under investigation in this study (≤7.5 mL venous blood or ≤400 μL capillary blood instead of the few drops required for MS and RDT) were more apparent when tests were administered by health professionals who were perceived as outsiders to the Ra-glai community. Individuals who were hesitant or even refused to participate in the study either (i) were uncomfortable with seeing blood and did not wish to endure the pain caused by blood sampling, (ii) perceived blood tests to be acceptable only when there were suspected malaria symptoms, or (iii) perceived the health worker as untrustworthy, e.g., due to negative experiences in previous inter-ethnic health encounters, or unfair compensation to the hardship the patient endured, e.g., loss of blood, loss of farming work, long traveling and transportation costs. General concerns were expressed about the high number of malaria surveys targeting Ra-glai for blood sampling that were conducted in the area, as people reported not always understanding the purpose of these procedures/studies or receiving information about what happened to their blood samples after the collection. In the health facility-based study, hesitancy toward blood sampling was addressed by involving village health workers, local leaders, and trusted individuals to sensitize the population to the study. 

### 3.4. Factors Determining the Use of Appropriate Malaria Treatment 

#### 3.4.1. Provider Challenges with Radical Cure 

At the time of the study, there was no specific guidance on radical cure to prevent the reoccurrence of malaria amongst vivax patients nor any systematic tools for local health staff to record any recurrent parasitaemia. Health staff had to rely on memory of whether an individual patient had had malaria previously. CHC staff followed the same diagnostic testing (RDT and MS) and treatment guidelines (Appendix A, Table 1, Table 2, Table 3, Table 4 and Table 5) as newly infected cases for recurring malaria cases. Health staff were also lacking information about the diagnostic options to test for G6PD deficiency, its benefits for the patient, and its importance for the prescription of PQ. The common procedures for vivax patients were such that after three days of observed CQ and PQ administration at the health facility, patients were asked to continue the rest of the PQ treatment regimen at home (Appendix A, Table 3 and Table 5). The administration of PQ at either the CHC and the DHC was said by staff not to be a problem as most patients complied, except for a few cases of young children who were unable to swallow the pills or vomited due to its bitter taste. Despite guidance on handling side effects caused by PQ by the national guidelines (Appendix A), health staff additionally faced understaffing and did not have the capacity or the designated function to ensure adherence to 14-day PQ. The administration of the rest of the PQ regimen at home was considered challenging, especially when it concerned young children (due to fears of side effects) and men who had plantation and forest work (due to forgetting to take their medication). 

Several healthcare professionals mentioned they were aware of possible adverse events caused by PQ, such as yellow skin, dark urine, acute fatigue, and vomiting. Negative accounts of perceived serious side effects following PQ administration were shared and led some health workers to be hesitant in prescribing PQ. The lack of G6PD diagnostic tools in routine diagnostic service combined with the lack of capacity to handle side effects caused by PQ implied enormous pressure on the prescriber. Serious side effects occasionally occurred in patients, and in such cases, the prescriber had to refer the patient to the district or provincial hospital, which added financial and psychological stress for the patient and his/her relatives. Patient complaints about serious side effects were perceived to cause damage to the prescriber’s reputation and to affect patients’ trust in the individual prescriber, leading again to reservations and concerns in prescribing PQ to patients. 

In the daily practice of CHCs, only the malaria officer was knowledgeable of different brand names and the dosages to treat different species of malaria. The rest of health staff had to consult (in person or by phone) the malaria officer for instructions if that person was off duty. The dependence on the malaria officer amongst health staff resulted in longer waiting times for patients and further fostered patients’ perceptions on trustworthy versus less trustworthy prescribers. These other health staff expressed uncertainty, even anxiety, when prescribing malaria treatment according to the national treatment guidelines [33], because they were not confident in calculating the daily dose of antimalarials using the age or weight of the patient (Appendix A). This fear was frequently instigated by the fact that patients often did not know their exact age, whilst in some cases, the age on their ID clearly did not match their observed age, which meant that patients’ vulnerability could not be assessed. Health workers also shared a concern that both Ra-glai adults and children had lower weights than the standard weights due to poor diet and harsh working/living conditions, which could make the patient at risk of side effects, particularly those caused by PQ. 

#### 3.4.2. Trust in Healthcare Providers 

Trust in health staff was key, particularly when it came to patients’ adherence to the prescribed course of medicine. People knew they had the disadvantage of having limited knowledge of diseases and treatments, and specific prescribers who were able to provide perceived effective treatment gained trust amongst not only the patient but also the community via word-of-mouth. High acceptance of medication to treat asymptomatic malaria was linked to people’s trust in the public provider who was perceived to make medical decisions in the patient’s best interest. Trust in the public provider was also linked to the long history of free-of-charge medical care including malaria diagnoses and treatment provided by the government to Ra-glai. Trust in public health staff was reflected in the increasing number of visitors at the CHC when specific prescribers were present. Some patients chose to wait for their preferred prescribers until they were available or revisited the CHC another day. Informants said they knew about the working schedule of specific CHC nurses and shared this information through the village’s social network. Some of them organized shared transportation so several people could visit the CHC on the day their preferred nurse was working. Trust was gained not only by prescribing effective treatments but also simply by being of Ra-glai ethnicity and being able to explain to patients in Ra-glai language about the expected outcomes, dosage, and the trajectory of antimalarials working against illness symptoms. Reported causes for mistrust included the lack of communication with Ra-glai patients by health staff from Kinh ethnicity, their differential treatment of Ra-glai patients compared to Kinh patients, and sometimes even discriminative language or gestures toward Ra-glai patients. 

#### 3.4.3. Inter-Ethnic Health Encounters 

We observed that most communication in the public health sphere occurred in Vietnamese (the language of the Kinh), which was difficult for Ra-glai patients to fully understand, especially when it concerned medical terms and concepts. This language barrier was said to further contribute to the perceived hierarchy between Kinh and Ra-glai and between doctors and patients. Constrained communication and misunderstandings occurred and were often perceived as the doctor expressing superiority and being inattentive to the patient’s needs. For instance, CHC staff expected patients to respect their official working hours but due to farming requirements and distance, patients could only come to CHCs outside of working hours. CHCs are mandated to treat uncomplicated medical problems, whilst severe cases are referred free-of-charge to the hospital by an ambulance. Such referral was interpreted by some patients as a sign of not caring or causing additional burdens to the patient, e.g., increased indirect treatment costs, since at least one family member would accompany the patient. Slow arrival of the ambulance and attitudes of the Kinh provider toward Ra-glai patients added more stress to the patient and further demonstrated differential treatment. 

## 4. Discussion

In this study, we assessed the social factors related to providing accurate diagnosis and effective treatment for vivax malaria in south-central Vietnam. The application of the novel G6PD diagnostic tools under investigation in this study is challenged by the absence of G6PD testing as part of routine diagnostics prior to the prescription of PQ and inadequate laboratory capacity and training of local health workers. The acceptance of the novel G6PD testing is additionally challenged by inter-ethnic trust issues and negative patient experiences at health facilities. 

Firstly, from an epidemiological perspective, the primary challenge for eliminating *P. vivax* is the large parasite reservoir of asymptomatic individuals [58,59], who are undetected by currently used diagnostics. The use of MS and RDT for the diagnosis of *P. vivax* is suboptimal because of varied quality of microscopy service and limited sensitivity of RDTs to detect low level parasitaemia and hypnozoite carriage [60,61,62]. Malaria elimination strategies in low transmission settings such as the GMS need highly sensitive tests for detecting low parasite density, including sub-microscopic infections, and better point-of-care diagnostics for G6PD deficiency to support the safe and effective radical cure of the parasite [28,32]. From a bio-technological perspective, the design of novel diagnostics for malaria must take into account not only the technical aspects, such as the threshold, specificity, and sensitivity of the test, but also the “social life” of these health technologies [63]. The “social life” of G6PD testing is understood as the making, assigning, and interpretation of meanings and rationality of this novel diagnostic tool amongst different actors including policy makers, malaria control program managers, health workers, patients, and community members. A better understanding of these features will provide insights into how to improve the design of G6PD tests as well as the communication to different actors on the benefits of the novel diagnostic test. In addition, in countries in the pre-elimination phase, where malaria prevalence is low and the capacity of the local health system is varied, it is critical for future interventions and malaria elimination programs to determine the relevant level of care that provides G6PD testing and radical cure, whether it will be at the village, commune, or district level. In many of these settings, the disproportionate concentration and the persistence of malaria transmission amongst ethnic minorities, mobile migrants, and forest workers despite the implementation of standardized interventions and existing diagnostic tools highlight the importance of accounting for contextual factors. We observed that when G6PD testing was provided free-of-charge and with explanations about what it was for, the study still faced a challenge to obtain the acceptance and the participation by some patients. This implies the importance of further studies in inter-ethnic trust, trust in public health service, and how community engagement could help improve the uptake of malaria elimination interventions. A critical approach to technologies [63] should include an in-depth understanding of the actor network, such as the power relations between different ethnicities and the doctor and the patient and how these influence the acceptance of the use of new diagnostic service. This understanding will inform the identification of changes needed at the policy and malaria elimination program level for the successful integration of these novel tools into clinical practice. 

Secondly, *P. vivax* causes relapses [10,64] and a socio-economic burden on the patient [4,5,6,7], although it generally presents with less severe illness than *P. falciparum.* As a result, patients with mild fever do not always actively seek malaria diagnosis and treatment at public health facilities. In this ethnic minority setting in Vietnam, taking a family member to a far-away health facility is detrimental to the demands of farming, and being absent from the farm for only a few days could jeopardize an entire family who are at risk of losing their year’s work and income. Many would rather explore a treatment option involving less traveling and work disruption when available. This flexible health-seeking behavior should be understood as people’s strategy to minimize the indirect cost associated with formal medical care, and behavior change communication should not emphasize this as “bad” or “deviant” behavior to avoid the stigmatization and perpetuation of existing stereotypes on poor and “backward” ethnic minorities [65,66]. Health workers should be provided with information and training on G6PD deficiency as well as the importance of adherence to be able to carry out radical cure. Decentralization of malaria diagnostics and management is critical to elimination strategies and yet presents huge challenges when servicing ethnic minorities or other forest workers with complex mobility patterns [35,37,38]. 

In the remaining hotspots in the GMS, healthcare providers need to anticipate the impact of swidden farming and mobility on delayed health-seeking behaviors [36,37,38,40,58] and assume that patients navigate uncertainties [67] and re-evaluate [35] different treatment options and providers as an adaptive strategy to mobility and farming demands. This context also includes the wide availability of OTC medicines in poor rural areas that are often unregulated, substandard and of unknown origin [51]. Evidence also suggests the limited utilization of public health service amongst ethnic minorities, even when it is free of charge [68,69,70,71]. Decentralized health services at the DHC and the CHC as well as village health workers face understaffing and inadequate laboratory capacity to handle the increasing dominance of *P. vivax* and associated challenges of ultra-sensitive diagnosis, G6PD deficiency, and radical cure prescription. The identification of appropriate providers for G6PD testing along with the needed investment in facilities and capacity building, including the training and sensitization of healthcare workers at local levels on G6PD deficiency and the benefits of testing for it in order to prescribe optimal radical cure to patients, will be essential to ensure quality service. Ra-glai patients’ trust in the public provider was a key determinant in both the acceptance of additional blood sampling for G6PD testing and patient adherence to PQ treatment. Therefore, investments must be made in improving communication between doctors and patients, ideally by offering an explanation of the diagnosis and treatment in local language to patients. Local health facilities can involve village health workers as translators when language barriers challenge the therapeutic encounter. Formative research, including community participatory techniques, can help initiate a dialogue with community members. They can participate in developing culturally sensitive messages and identify relevant communication channels and trusted individuals to disseminate health messages to their peers. 

The next challenge for rolling out G6PD testing and radical cure is how to communicate these complex concepts to populations whose conceptualization of malaria is different from the biomedical explanation and who also lack the words that would allow accurate translation of the biomedical concepts [36,37,40]. Inter-ethnic encounters at health facilities, whether for malaria studies or routine testing and treatment, leave an impact on patients and shape their future perceptions of and trust in medicine in general. Our study highlights that involving local health professionals, village health workers, and respected leaders from the local community in the study design and the implementation helps narrow these conceptual differences and its impact on trust. Research challenges such as drop-out, survey fatigue, and hesitancy toward blood sampling can be met by meaningful dialogues and discussions between the researcher and the local leaders including traditional leaders, authorities, and community members. Sensitizing communities to the acceptance and the utilization of new malaria tests that require additional blood sampling and the interpretation and the implication of all diagnostic results should be done by trusted individuals and in the language of the targeted ethnicity. Malaria programs and study teams should consider a feedback-loop mechanism that provides correct information on malaria diagnoses and treatment so people could make informed decisions. 

## 5. Conclusions

This mixed-method study demonstrates the importance of understanding the local context of malaria that will facilitate the public acceptance of novel diagnostic tools and the uptake of public health interventions, which is crucial to the elimination of malaria from Vietnam by 2030. The diagnostic and treatment components of the malaria elimination strategy will need to address not only the technical and logistical aspects of the new G6PD diagnostic tool but also challenges related to the health system, service delivery capacity, and the patient’s demands and preferences. Improved diagnostic tools are an essential component of the elimination agenda provided that sufficient linkages with other strategies for vector control, treatment, surveillance, and community participation are secured. 

## Figures and Tables

**Figure 1 pathogens-10-00026-f001:**
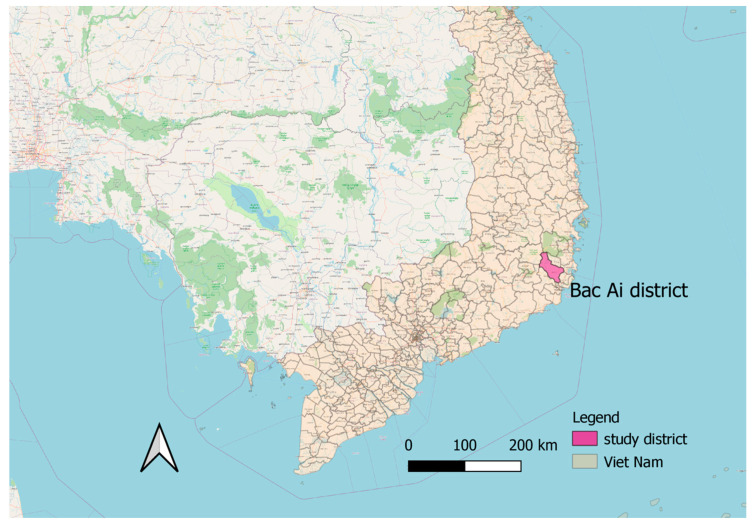
Map of the study district, Bac Ai District, Ninh Thuan Province, Vietnam [52]. Scale bar = 200 km.

**Table 1 pathogens-10-00026-t001:** Respondent characteristics from the quantitative strands.

	Frequency (n)	Proportion (%)
**Household representatives included in the cross-sectional survey (N = 202)**		
Sex		
- Male	91	45.1
- Female	111	54.9
Median of age (range), IQR	40 (14; 73); IQR (18; 68)	
Ethnicity		
- Ra-glai	199	98.5
- Kinh	3	1.5
Religion		
- Christianity	5	2.5
- Animism	196	97.0
- Buddhism	1	0.5
Occupations (multiple replies possible)		
- Sedentary farming	149	73.8
- Rotational farming	174	86.1
- Seasonal plantation work	137	67.8
Median of days being away from home for work (n = 180)	6 (min = 1; max = 90); IQR (1; 60)	
**Patients enrolled in the health-facility based survey (N = 176)**		
Sex		
- Male	62	35.2
- Female	114	64.8
Ra-glai ethnicity	176	100
Median of age (range, IQR)	30 (2; 77), IQR (5; 64)	

**Table 2 pathogens-10-00026-t002:** Knowledge of malaria in both quantitative strands.

	n	%
**Household representatives included in the cross-sectional survey (N = 202)**		
Has not heard about different types of malaria	153	75.7
Knows about ‘hidden’ (asymptomatic) malaria	44	21.8
Knows about the possibility of relapses	60	29.7
Did not think a person could have malaria without having symptoms	105	52.0
Perceived causes of relapse (n = 60)		
- Did not complete medicine	4	6.7
- Incorrect treatment	2	3.3
- Drinking dirty water	1	1.7
- Going/sleeping in the forest	22	36.7
- Staying in the old village	10	16.7
- Polluted environment	1	1.7
- Mosquito bites	21	35.0
- Parasites (“*ký sinh trùng*”, “*ana hula*”, “*sâu*”)	29	48.3
- Weak body	1	1.7
- Do not know	12	20.0
**Patients enrolled in the health-facility based survey** **(N = 176)**		
Positive malaria results (using conventional RDT)	8	4.6
Positive for *P. vivax* (using microscopy)	2	1.1
Has heard about the two common types of malaria (vivax and falciparum)	10	5.7
Thinks that each type of malaria has different symptoms	7	4.0
Thinks that falciparum causes more severe symptoms	3	1.7
Suspects their fever was caused by malaria	21	11.9
Patients who report to have been infected with malaria in the past	7	4.0
Patients who suspected their current fever was caused by a relapse of that past malaria episode (n = 7)	1	14.3

**Table 3 pathogens-10-00026-t003:** Perceptions of malaria aetiology in the community-based cross-sectional survey (N = 202).

	Free Response		Prompted Response	
	n	%	n	%
Reported common fevers				
- Viral fever	3	1.5	18	8.9
- Influenza	122	60.4	195	96.5
- Malaria	64	31.7	168	83.2
- Fever caused by pollution	1	0.5	173	85.6
- Fever caused by the weather	0	0	7	3.5
- Fever caused by contaminated food	0	0	32	15.8
- Dengue fever	7	3.5	1	0.5
- Other (diarrhoea, fatigue, stomach pain)	5	2.5	37	18.3
- Do not know	48	23.8	7	3.5
- No answer	1	0.5	9	4.5
Perceived causes of malaria (multiple choice)				
- Drinking dirty water	1	0.5	33	16.3
- Going/sleeping in the forest	91	45.1	180	89.1
- Being in the rain	4	2.0	14	6.9
- Staying in the old village	44	21.8	157	77.7
- Hot weather	0	0	0	0
- Polluted environment	2	1.0	37	18.3
- Poor personal hygiene	0	0	14	6.9
- Dirty house	0	0	3	1.5
- Virus/viruses	1	0.5	1	0.04
- Mosquito bites	148	73.3	181	89.6
- Parasites (“*ký sinh trùng*”, “*ana hula*”, “*sâu*”)	1	0.5	43	21.3
- Insects in the forest/water	4	1.2	59	28.2
- Other (working away from home, not sleeping under a net, climate change)	3	1.5	3	1.5
- Do not know	32	15.8	26	12.9
- No answer	1	0.5	1	0.5
Perceived symptoms of malaria (multiple choice)				
- Fever	146	72.6	192	95.5
- Dizzy	3	1.5	100	49.8
- Chills	130	64.7	189	94.0
- Headache	26	12.9	171	85.1
- Body pain	5	2.5	58	28.9
- Insomnia	3	1.5	27	13.4
- Vomit	1	0.5	20	10.0
- Stiffness	2	1.0	13	6.5
- Lack of appetite	0	0	10	5.0
- Nausea	0	0	19	9.5
- Fatigue	20	10.0	128	63.7
- Unable to work	0	0	1	0.5
- Diarrhoea	0	0	1	0.5
- Hot body	9	4.5	26	13.0
- Difficult indigestion	0	0	1	0.5
- Cold body	8	4.0	27	13.4
- Other (coughing, sweat, yellow skin)	0		4	2.0
- Do not know	40	19.9	13	6.5
- No answer	1	0.5	0	0

**Table 4 pathogens-10-00026-t004:** Perceptions of malaria treatment in community-based cross-sectional survey (N = 202).

	n	%
Thinks a person can be infected with malaria whilst showing no symptoms (asymptomatic infection)		
- No	105	52.0
- Yes	44	21.8
- Do not know	52	25.7
- No answer	1	0.5
Perceived the necessity of treatment for asymptomatic infection (given being prescribed by a health worker)		
- Not necessary	2	1.0
- Necessary	190	94.1
- Do not know	9	4.5
- No answer	1	0.5
Reported importance of adherence to treatment (N = 202)		
- Patients should continue taking all medicine even when better	185	91.5
- Patients should stop taking medicine when feeling better	10	5.0
- Do not know	6	3.0
- No answer	1	0.5
Use of malaria treatment during last self-reported malaria episode (N = 21)		
Sought treatment from CHC first	21	100
Received treatment from CHC during last malaria episode	19	90.5
Received different types of pills during last malaria episode	3	14.3
Reported treatment regime		
Do not remember	10	47.6
1 day	4	19.0
3 days	3	14.3
5 days	3	14.3
10 day	1	4.8
Received supplements to malaria treatment	14	66.7
Types of supplement (n = 14, multiple responses)		
- Injection	1	7.1
- IV drips	13	92.9
- Spiritual sacrifice	2	14.3

**Table 5 pathogens-10-00026-t005:** Perceptions of malaria diagnosis in the community-based cross-sectional survey (N = 202).

	n	%
Self-reported past malaria infection	21	10.4
Malaria diagnosis at health facility during last malaria episode (n = 21) (multiple choice)		
RDT	21	100
RDT, then microscopy	12	57.1
RDT, then unknown	4	19.0
Median days from onset malaria symptoms to seeking diagnosis and treatment	3 (min = 1; max = 4), IQR (2; 4)	

## Data Availability

The authors confirm that the data supporting the findings of this study are available within the article and its Appendix A.

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
