# Peer review of "Diagnostic Practices and Treatment for P. vivax in the InterEthnic Therapeutic Encounter of South-Central Vietnam: A Mixed-Methods Study"

_pathogens, 2020, doi:10.3390/pathogens10010026_

Round 1
Reviewer 1 Report
- conext : As we are talking about the diagnostic test. I ask the authors to insert a chapter on the current test used in the study area. Also to specify the difficulties associated with its use in the general population. In the method, the authors speak of its absence but the possibility of doing the test as part of the research. They must add this point to the justifications of the study.
- Method : The 22 selected houses refused, The authors explained the reasons for the refusals well. It would be interesting to explain sampling method. I think that in the method is harshly qualitative, the authors declare a mixed study . I did not find too much quantitative aspect, nor even univariate test.
- result : The results are very detailed and provide sufficient information. They are allowed to meet the objective of the study.
I propose to the authors to add the number of positive vivax tests and the number of positive G6PD.
Author Response
Point 1
Context: As we are talking about the diagnostic test. I ask the authors to insert a chapter on the current test used in the study area. Also to specify the difficulties associated with its use in the general population. In the method, the authors speak of its absence but the possibility of doing the test as part of the research. They must add this point to the justifications of the study.
Response 1
We thank and agree with the reviewer’s comment.
As we explained in the revised manuscript, even though the national guideline recommends G6PD testing for haemolysis cases after treatment has not provided the specifications of the G6PD deficiency testing methods after administering antimalarial treatment. In practice, G6PD deficiency diagnostics is limited to a number of specialized hospitals at national, regional and provincial level (with testing methods varied including both qualitative and quantitative) but unavailable at district or commune level. In our study, the larger health facility-based study examined the application of a novel quantitative diagnostic tool G6PD RDT Accessbio / Carestart, USA, which required the participants to give either £7.5ml venous blood or £400ml capillary blood to undertake different malaria tests using methods and tests as detailed in the approved study protocol. We intend to publish findings separately.
In response we have revised the manuscript under on page 2, Introduction, lines 78 to 84:
“In line with the WHO’s treatment guidelines, the NMCP in Vietnam recommends the use of PQ-based radical cure for patients with uncomplicated P. vivax and P. ovale malaria [33]. Whilst the national treatment guidelines do not recommend G6PD testing prior to administering primaquine they recommended patients with significant haemolysis following primaquine although do not specify the testing method. The availability of G6PD deficiency diagnostics is limited to a number of specialized hospitals at national, regional and provincial levels (with testing methods varied, including both quantitative and qualitative) but unavailable at district or commune level.”
And also on page 3 (lines 151 to 157).
“At the time the study was conducted, malaria diagnostics used by the local health services included finger prick sampling, microscopic examination of giemsa stained blood films and a conventional rapid diagnostic test (RDT). Although routine G6PD testing was not part of healthcare, some health centers in Bac Ai district participated in G6PD research studies using the qualitative fluorescent spot test (FST) [48]. Previous malaria studies conducted in the region reported no challenges on blood sampling and the acceptance of malaria diagnostic tools amongst the local population [46–48].”
Point 2
Method: The 22 selected houses refused. The authors explained the reasons for the refusals well. It would be interesting to explain sampling method. I think that in the method is harshly qualitative, the authors declare a mixed study. I did not find too much quantitative aspect, nor even univariate test.
Response 2
We thank you for your comment on the method. The sample was a random selection using the using a computer program (http://www.random.org).
We have revised the wording of methods accordingly page 5, lines 247-250.
“Sampling. In the cross-sectional survey, 892 households from four villages (based on the local census data of Bac Ai) were used as the sampling frame, of which 202 were selected randomly using a computer program (http://www.random.org).”
On the second part of the reviewer’s comment on quantitative analysis, we would like to explain why we have decided not to present the univariable analysis despite this being in the initial analysis plan - for exploring the association between the clinical outcome treatment adherence. However the survey shows that only 21 out of 202 patients with malaria who self-reported malaria and only three of these had had malaria in the past 12 months. The application of a formal univariable test was therefore not relevant, and the basic descriptive analysis is more informative.
Point 3
Result: The results are very detailed and provide sufficient information. They are allowed to meet the objective of the study. I propose to the authors to add the number of positive vivax tests and the number of positive G6PD.
Response 3
We thank the reviewer for providing us with both encouraging and constructive comments. We have revised the manuscript on page 7, line 322 to 324.
“A total of 4.6% (8/176) of enrolled patients in the health-facility based study had a positive malaria test, according to the conventional RDT, of which 2 infections were confirmed by microscopy to be P. vivax (25.0% of total number of malaria cases).”
We have also included a row in the table on line 327 to report on the proportion of P. vivax patients. The results of G6PD deficiency testing using the novel diagnostic tool will be reported separately, as stated on page 4, line 202.

Reviewer 2 Report
Authors conducted a mixed-method study to explore the factors that affect acceptability of G6PD testing, treatment seeking behaviors and adherence to current regimens. This topic is relevant to achieve Plasmodium vivax elimination. Study design is appropriate. The authors focused on ethnic minority population and demonstrated that a trust between health service providers and the patients would be important.
To improve the manuscript, I recommend some minor revisions as follows. The national malaria control programs (NMCP) in Vietnam recommends G6PD testing to guide PQ-based radical cure. However, most of endemic provinces have not yet been ready for performing the G6PD testing and the PQ treatment. The authors should explain the reason why there is such gap between the NMCP and the provinces, and how to overcome this issue.
The authors mentioned that some patients hesitated to collect blood sample for additional test (G6PD testing), even they accepted for taking blood sample for malaria RDT or microscopy. Please describe how many µL of blood are needed for G6PD testing. If the volume is small, blood sampling can be done by finger-prick by lancet that is easy and less invasive.
China is member country of GMS however it is specifically only Yunnan Province and the Guangxi Zhuang autonomous region. The authors should clarify this point otherwise some readers might be confused.
Author Response
Response to Reviewer 2
Point 1
Authors conducted a mixed-method study to explore the factors that affect acceptability of G6PD testing, treatment seeking behaviors and adherence to current regimens. This topic is relevant to achieve Plasmodium vivax elimination. Study design is appropriate. The authors focused on ethnic minority population and demonstrated that a trust between health service providers and the patients would be important.
Response 1
We thank the reviewer for these encouraging comments.
Point 2
To improve the manuscript, I recommend some minor revisions as follows. The national malaria control programs (NMCP) in Vietnam recommends G6PD testing to guide PQ-based radical cure. However, most of endemic provinces have not yet been ready for performing the G6PD testing and the PQ treatment. The authors should explain the reason why there is such gap between the NMCP and the provinces, and how to overcome this issue.
Response 2
We thank and agree with the reviewer’s comment. In response to the reviewer’s suggestion, we have included a paragraph on page 2, lines 82-86 to highlight the gap in the national technical guideline and implementation capacity for G6PD testing.
“The availability of G6PD deficiency diagnostics is limited to a number of specialized hospitals at national, regional and provincial levels (with testing methods varied, including both quantitative and qualitative) but unavailable at district or commune level. Hence most endemic provinces have not yet acquired the necessary diagnostic capacity or equipment to provide this additional health service, and this undermines the safe and effective treatment of P. vivax malaria.”
Point 3
The authors mentioned that some patients hesitated to collect blood sample for additional test (G6PD testing), even they accepted for taking blood sample for malaria RDT or microscopy. Please describe how many µL of blood are needed for G6PD testing. If the volume is small, blood sampling can be done by finger-prick by lancet that is easy and less invasive.
Response 3
We agree with the comment from the reviewer and have revised the manuscript on page 5, line 252-254.
“The study collected £7.5ml venous blood from one randomly selected household member and £400ml capillary blood each of the rest of household members above the age of one year (data on venous participants will be presented separately).”
And also on page 12, line 518 to 520:
“Reservations towards the additional blood sampling needed for the G6PD test under investigation in this study (£7.5ml venous blood or £400ml capillary blood instead of the few drops required for MS and RDT),”
Point 4
China is member country of GMS however it is specifically only Yunnan Province and the Guangxi Zhuang autonomous region. The authors should clarify this point otherwise some readers might be confused.
Response 4
We thank the reviewer for the comment and have specified this on page 2, line 53 and 54 in the revised manuscript.
“Six countries in the Greater Mekong Sub-region (GMS), including Cambodia, Yunnan province and the Guangxi Zhuang Autonomous Region of China, Laos, Myanmar, Thailand and Vietnam, have committed to eliminate malaria by 2030” [8,9].
